# More than 50% of Persistent Myocardial Scarring at One Year in “Infarct-like” Acute Myocarditis Evaluated by CMR

**DOI:** 10.3390/jcm10204677

**Published:** 2021-10-13

**Authors:** Thibaut Pommier, Thibault Leclercq, Charles Guenancia, Simon Tisserand, Céline Lairet, Max Carré, Alain Lalande, Florence Bichat, Maud Maza, Marianne Zeller, Alexandre Cochet, Yves Cottin

**Affiliations:** 1Department of Cardiology, CHU Dijon-Bourgogne, 14 rue Gaffarel, CEDEX, 21079 Dijon, France; thibault.leclercq@chu-dijon.fr (T.L.); charles.guenancia@chu-dijon.fr (C.G.); celine_gc26@hotmail.fr (C.L.); max.kenzo.carre@gmail.com (M.C.); florence.bichat@chu-dijon.fr (F.B.); maud.maza@chu-dijon.fr (M.M.); marianne.zeller@chu-dijon.fr (M.Z.); yves.cottin@chu-dijon.fr (Y.C.); 2Department of MRI, CHU, 21000 Dijon, France; simon.tisserand@chu-dijon.fr (S.T.); alain.lalande@u-bourgogne.fr (A.L.); alexandre.cochet@u-bourgogne.fr (A.C.); 3Laboratory ImVIA, EA 7535, University of Burgundy, 21000 Dijon, France

**Keywords:** acute myocarditis, cardiac magnetic resonance, one year follow-up

## Abstract

Background: Cardiac magnetic resonance (CMR) has emerged as a reference tool for the non-invasive diagnosis of myocarditis. However, its role in follow-up (FU) after the acute event is unclear. The objectives were to assess the evolution of CMR parameters between the acute phase of infarct-like myocarditis and 12 months thereafter and to identify the predictive factors of persistent myocardial scarring at one year. Methods: All patients with infarct-like acute myocarditis confirmed by CMR were included. CMR was performed within 8 days following symptom onset, at 3 months and at one year. One-year FU included ECG, a cardiac stress test, Holter recording, biological assessments, medical history and a quality-of-life questionnaire. Patients were classified according to the presence or absence of complete recovery at one year based on the CMR evaluation. Results: A total of 174 patients were included, and 147 patients had three CMR. At one year, 79 patients (54%) exhibited persistent myocardial scarring on CMR. A multivariate analysis showed that high peak troponin at the acute phase (OR: 3.0—95%CI: 1.16–7.96—*p* = 0.024) and the initial extent of late gadolinium enhancement (LGE) (OR: 1.1—95%CI: 1.03–1.19—*p* = 0.006) were independent predictors of persistent myocardial scarring. Moreover, patients with myocardial scarring on the FU CMR were more likely to have premature ventricular contractions during the cardiac stress test (25% versus 9%, *p* = 0.008). Conclusion: Less than 50% of patients with infarct-like acute myocarditis showed complete recovery at one year. Although major adverse cardiac events were rare, ventricular dysrhythmias at one year were more frequent in patients with persistent myocardial scarring.

## 1. Introduction

Myocarditis is an inflammatory disease with a variable course, ranging from complete remission to severe complications [1]. Myocarditis remains a difficult diagnosis because of the various clinical presentations: infarct-like syndrome, acute or chronic heart failure, cardiac arrhythmia or cardiogenic shock [2]. However, chest pain mimicking acute myocardial infarction is the most frequent presentation. The most common cause of myocarditis is viral infection, other causes comprise systemic or toxic diseases [3].

According to previous studies, the prognosis of this disease is rather good with spontaneous clinical healing in many patients. However, complications may occur such as sudden cardiac death (SCD) [4] or dilated cardiomyopathy (DCM) [5] and myocarditis remains in young patients a major cause of deaths.

Cardiovascular magnetic resonance (CMR) is now a major diagnostic tool for the non-invasive assessment of myocardial inflammation and has emerged as a reference for the diagnosis of myocarditis [6], thus replacing highly invasive endomyocardial biopsy, which is now used for specific cases [7]. Today, CMR is more readily available and is the first-line diagnostic test in many centers.

However, the interest of CMR in evaluating the prognosis of acute myocarditis remains unclear [8]. There have been few specific studies on the CMR follow-up (FU) of myocarditis, and most of these were limited to an evaluation at 3 or 6 months. Yet, one known indicator of a poor outcome in myocarditis is persistent late gadolinium enhancement (LGE) on CMR [9]. Nonetheless, the results of follow-up studies on infarct-like myocarditis are contradictory as some report an association with major adverse cardiac events while others report the absence of cardiac complications.

The objectives of our study were thus to assess, in patients with LGE-positive infarct-like myocarditis:(1)The evolution of CMR parameters between the acute phase and 12 months thereafter.(2)The predictors of persistent myocardial scarring at one year and the long-term prognosis of the infarct-like form.

## 2. Methods

### 2.1. Study Flow Chart

All cases of infarct-like acute myocarditis, revealed as an acute coronary syndrome with chest pain, elevated troponin levels and ECG changes, from April 2012 to April 2019, were included in a prospective single-center study at Dijon University Hospital. The diagnosis of myocarditis was based on clinical symptoms and was confirmed by CMR. The study sample consisted of 174 patients, with at least one-year follow-up for 147 of them. All of the patients gave their consent to enter the protocol in this clinical study using completely anonymized data.

From the time of admission and for each patient hospitalized with acute myocarditis, demographic data, vascular risk factors, past medical history and biological assessments were collected. Myocarditis management in term of drugs was left to the physician’s discretion.

Cardiac magnetic resonance (CMR 1) was performed within 8 days following symptom onset in order to validate the diagnosis of myocarditis with the presence of typical non-ischemic late gadolinium enhancement (LGE). CMR was also performed at 3 months (CMR 2) and one year (CMR 3) after the acute event. One-year FU included ECG, a cardiac stress test, Holter recording, biological assessments, medical history and a quality-of-life questionnaire. After the CMR at one year, patients were divided into two groups according to the presence or absence of LGE. The flow chart is shown in Figure 1. The CMR protocol is defined in the Appendix A. Major adverse cardiac events (MACE) were defined as cardiac death (death from all cardiac causes, including SCD and heart failure) and serious rhythm disorders (ventricular tachycardia or fibrillation).

### 2.2. CMR Analysis

All CMR images were analyzed by two experienced practitioners using Syngo.via software. The ventricular volume and function were measured for the left ventricle (LV) using standard techniques. Short and long axis cine-sequences images were used for the evaluation of LV function, LV volume and LV mass: the endocardial and epicardial borders were manually drawn on the end-diastolic and end-systolic short-axis. LV end-diastolic volume (LVEDV), LV end-systolic volume (LVESV), ejection fraction (EF) and LV mass were determined using these outlines. In our study, the distribution and location of LGE in the myocardium were used to diagnose myocarditis (typical non-ischemic LGE affecting the subepicardial and mid areas regions).

The extent of myocardial involvement was quantified using a simplified visual quantitative score (SQS) of LGE [10] with the 17-segment LV division of the American Heart Association, which has already been employed in several studies about myocarditis. Segments with LGE were scored according to the number of quartiles involved (1–25%, 26–50%, 51–75% and 76 to 100%). The CMR results were finally expressed as the percentage of LV myocardium involved calculated by adding the 17 segment scores (LGE extent = n% VG = (addition of 17 segment scores/68) × 100). Each case was also classified according to the predominant location of LGE: infero-lateral, antero-septo-apical or diffuse. All parameters were measured for each of the three CMR. Intra- and inter-observer reliability were checked and there was a difference of less than 5% between the two measurements proving the reproducibility of the measurements (visual quantitative score and predominant location of myocarditis).

### 2.3. Statistical Analysis

Statistical analyses were performed using SPSS version 12.0.1 (Statistical Package for the Social Sciences, IBM Inc, Armonk, NY, USA). Dichotomous variables were expressed as *n* (%) and continuous variables as median [interquartile range]. A Kolmogorov–Smirnov test was performed to analyze the normality of continuous variables. A Mann–Whitney test (skewed data) or Student’s *t*-test (unskewed data) was used to compare continuous data, and the Chi 2 test or Fisher’s test was used for dichotomous data. The significance threshold was set at 5%. Continuous paired data were compared by Friedman test (3 variables) or Wilcoxon test (2 variables) and categorical paired data were compared by Cochran Q test (3 variables) or Mc Nemar test (2 variables). Multivariate logistic regression models were built to estimate lack of one-year recovery and including variables associated with the outcome in the univariate analysis. The inclusion threshold was set at 20%.

## 3. Results

### 3.1. Patients Baseline Characteristics

The baseline characteristics on admission to the hospital are summarized in Table 1.

The study included 174 patients, with a median age of 39. The majority of patients reported a recent history of infection or flu-like symptoms before the onset of the symptoms. Chest pain was present in all of the patients. Most patients had an abnormal ECG on admission (72%) with repolarization abnormalities as the most common finding (66%). The median peak troponin was 7.4 μg/L. At discharge from the hospital, nearly 90% of the patients had a prescription for a cardio-protective treatment comprising a beta blocker and an ACE inhibitor.

The baseline CMR examination was performed 3 ± 5 days after the onset of symptoms. Results are described in Table 2.

Cardiac systolic function was preserved in most patients. Twenty-seven patients (16%) had LV dysfunction (defined as LVEF < 50%). LGE was present in every patient and myocardial involvement was principally infero-lateral. The predominant location of LGE was antero-septo-apical in 14 (8%) patients, infero-lateral in 133 (77%) patients and diffuse in 27 (15%) patients.

The CMR results at baseline (CMR 1) and FU (CMR 2 and CMR 3) are summarized in Table 2.

At 3 ± 2 months, the LGE had resolved entirely in 40 patients (24%). A total of 147 patients returned for the follow-up CMR 12 months after the acute event. For one patient with an implanted cardioverter-defibrillator, the examination was performed using recent rhythmology technology. On the CMR analysis, LGE persisted in 79 patients (54%).

### 3.2. Follow-Up Results

The follow-up was complete for 147 patients after a period of 12 ± 3 months from the original diagnosis. One patient experienced cardiac death before the one-year CMR control and had precisely persistent myocarditis at three months, another patient died of cancer, and the others were lost to follow-up. One patient with LV dysfunction was fitted with an implanted cardioverter-defibrillator.

Patients were classified according to the presence or absence of complete recovery at one year, based on the CMR 3 evaluation (Figure 2). At one year, 79 patients (54%) exhibited persistent myocardial scarring on CMR, while the remaining 68 patients were considered healthy. The patients’ baseline characteristics in both groups are described in Table 3 and the CMR parameters at baseline in both groups are described in Table 4.

Regarding the CMR parameters, patients with persistent myocardial scarring at one year have a greater initial extent of LGE, slightly more dilated left ventricles and especially significantly higher global native T1 mapping and ECV compared to the patients with complete recovery at one year. Interestingly, most of the patients with complete recovery at one year already had complete recovery at 3 months. In contrast, patients with myocardial scarring at one year already had scarring at three months. The predominant location of LGE did not seem to be a predictor of recovery from the disease; however, nearly 90% of the patients with persistent myocardial scarring had an infero-lateral location.

In the multivariate analysis, only high peak troponin (OR: 3.0—95%CI: 1.16–7.96—*p* = 0.024) and the initial extent of LGE (OR: 1.1—95%CI: 1.03–1.19—*p* = 0.006) were baseline predictors of the absence of complete recovery at one year (Table 4).

In the one-year FU exams (Table 5), patients with persistent myocardial scarring on the follow-up CMR more often had premature ventricular contractions (PVC) during the cardiac stress test (more than five beats during the stress test) than did patients with complete recovery (25% versus 9%, *p* = 0.008). In most of the patients with cardiac stress PVC, the PVC originated from the site of the persistent inflammation on the CMR (about two-thirds). On the 24-h Holter monitor, the PVC burden was not different among groups with or without persistent myocardial scarring at one year. Moreover, no sustained ventricular arrhythmias and no atrial fibrillation were found on the cardiac stress test or the Holter recording. Concerning the FU 12-lead ECGs of the patients: the ECG was often normal and repolarization abnormalities remained the most frequent abnormalities.

At one year, there was no significant difference between the two groups regarding cardiac enzymes or inflammatory parameters (NT-pro BNP or C-reactive protein). Clinically, 31% of the patients remained symptomatic at one year (chest pain in 25% and dyspnea in 18% of patients), with no statistical difference between patients with and those without complete recovery on the CMR. Regarding the quality-of-life scores, no significant difference was found between the groups for any of the nine dimensions. In our cohort, the number of prior autoimmune diseases at baseline was 8%. Interestingly, 16% of patients had autoimmune diseases at one year, highlighting the discovery of some autoimmune diseases during the management of myocarditis.

## 4. Discussion

Our study is the first to evaluate, in a particular population of patients with infarct-like myocarditis, the evolution of CMR parameters (performed at the acute phase, 3 months and one year thereafter) coupled with clinical, biological and rhythmic parameters (Figure 3). We found that more than 50% of patients with infarct-like myocarditis evidenced the absence of complete recovery at one year. Moreover, we identified two independent predictors of persistent myocardial scarring: a higher troponin peak at the acute phase and more extensive LGE on the initial CMR. In addition, the study showed that patients with persistent myocardial scarring more often exhibited rhythm disorders during the stress test, and one patient experienced fatal ventricular rhythm disorder with LGE at the 3 months CMR.

### 4.1. Infarct-like Acute Myocarditis and CMR Findings

Few studies have examined the follow-up of patients with infarct-like myocarditis, and the results of such studies are contradictory. For example, Chopra et al. [11] reported that infarct-like acute myocarditis was more often associated with MACE than other forms, especially sustained ventricular tachycardia and the recurrence of myocarditis. In contrast, Caforio et al. [12] found that the long-term prognosis of the infarct-like form was better than that of myocarditis with arrhythmia or heart failure. In the same way, Danti et al. [13] reported no arrhythmia or further cardiac complications in their 23 patients with the infarct-like form; however, the follow-up was only two months. The results of Faletti et al. [14] were in line with the Danti study as all patients were free of symptoms with no arrhythmia or further cardiac complications at 6-months FU. The differences between the Chopra study and others could be related to differences in the duration of the follow-up: in the Chopra study, the adverse cardiac events occurred after the first year of follow-up concerning the infarct-like form.

Regarding LGE, some studies have demonstrated an association with an adverse prognosis [15].

In their study, Barone-Rochette et al. [10] reported an association between the LGE extent at 3 months and adverse cardiovascular outcomes. In 2012, Grün [15] conducted the largest study on LGE and outcomes in myocarditis (all forms) with 203 patients with biopsy-proven myocarditis, of whom 53% were found to have LGE on the baseline CMR. The authors reported that LGE was the best independent predictor of overall and cardiovascular mortality. None of the patients without LGE experienced SCD in a follow-up of 4.7 years, while 18 of the 108 patients (17%) with LGE experienced SCD.

In the ITAMY study [9] of patients with acute myocarditis (mainly infarct-like myocarditis) and preserved LVEF, LGE in the mid wall layer of the anteroseptal myocardial segment was associated with a worse clinical prognosis than other patterns of presentation. In comparison, our study has a follow-up comprising several CMR with a one-year examination (not present in the ITAMY study), and a rhythmic evaluation with a cardiac stress test representing a novelty in the studies focused on the prognosis of myocarditis. Moreover, Løgstrup BB et al. [16] showed recently that in patients with acute myocarditis, global longitudinal strain in echocardiography adds important information that can support clinical and conventional echocardiographic evaluation, especially in patients with preserved LV ejection fraction, highlighting the potential help of the novel technique in echocardiography in addition to CMR, especially if patients had complete recovery 3 months after the acute phase of myocarditis.

Finally, the precise determination of the composition of persistent LGE at one year would be interesting and remain a challenge, even though the appearance is compatible with fibrosis. Diagnostic accuracy is now further enhanced with novel T1 and T2 mapping approaches, and extracellular volume (ECV) quantification on CMR [17]. Moreover, native T1 mapping and ECV were associated with the presence of LGE at follow-up. Nevertheless, other methods such as using blood biomarkers would be a challenge for the future.

### 4.2. Clinical Implications

It is important to underline that most of our patients were young with limited cardiovascular risk factors, and evidenced typical infarct-like myocarditis with usually preserved LV function at the acute phase, thus highlighting the importance of our results on the management of this “otherwise healthy” population. The results of the present study may lead to modifications in the clinical management of acute myocarditis in routine practice.

The only factor other than the initial LGE extent to predict the persistence of myocardial scarring at one year was the troponin peak at the acute phase, stressing the need to monitor the troponin peak in this phase and to select those patients at risk in the follow-up. This association was not found by Berg [18] in his study, in which levels of cardiac enzymes at baseline did not predict changes in LGE at 3 months.

Moreover, one important result of our study concerns the evaluation of cardiac rhythm. Several studies have shown that myocarditis can potentially cause SCD. However, there are few data on the long-term rhythm risk in these patients. Indeed, the 2013 ESC guidelines do not recommend any specific follow-up [7]. In this study, cardiac monitoring showed that 10% of these patients had minor rhythm events in the acute phase, essentially PVC and NSVT. Then, patients with persistent myocardial scarring on the follow-up CMR were more likely than those without to present PVC during the cardiac stress test. This is the first study to report such results about rhythm disorders in a specific population of patients with infarct-like myocarditis, and treatment with beta blockers seems to be particularly suitable in this group of patients. For example, the indication of beta blockers could be based on the occurrence of PVC during cardiac stress tests in patients with persistent myocardial scarring to prevent life-threatening arrhythmias. Moreover, CMR at one year is also a trail to explore for a better identification of the patients at risk of complications because persistent LGE is associated with these rhythmic events. Nevertheless, we also noted that patients treated with aspirin tended to be less likely to recover at one year after the acute event. In animal models, the administration of high doses of aspirin seems to promote the progression of myocardial involvement with an increase in mortality, leading to a reflection on its use in myocarditis [19].

Finally, our study also found a larger proportion of patients with adverse cardiac remodeling (LV dilation) at one year than did previous studies, even though there was no statistical difference according to the presence or absence of LGE. This result underlines the need for optimized cardio-protective treatment, particularly the prescription of ACE inhibitors, in patients with infarct-like myocarditis, even though there is still no consensus on the duration of the therapy. Other studies including therapeutic trials are necessary to improve the management of myocarditis in the long term.

### 4.3. Limitations

Our study has some limitations. We did not carry out a histological validation for the diagnosis of myocarditis because our patients did not meet the clinical indications for endomyocardial biopsy, and it would not have been ethical to expose patients to this risk [20].

The diagnosis of acute myocarditis was based on the association of acute clinical symptoms associated with a typical pattern of LGE on CMR, with other signs consistent with acute myocarditis (myocardial edema on T2-weighted images, pericardial effusion, LV dysfunction).

Coronary angiography was not performed if the clinical presumption of myocarditis was very high (young age, recent viral history, inflammatory syndrome) and if the patient did not have an indication for urgent coronary angiography (haemodynamic instability, chest pain refractory, acute heart failure or life-threatening arrhythmias). In addition, easy access to CMR in our center always allowed a rapid confirmation of the diagnosis compared to a lot of other hospitals in France, and no patients had any myocardial infarction on CMR. We wanted to avoid the risk of an invasive examination in a population at low risk of coronary artery disease, especially if there is a significant biological inflammatory syndrome. Indeed, in case of doubt, a cardiac CT was systematically performed.

We did not perform early gadolinium-enhanced (eGE) images as recommended by the Lake Louise Criteria (LLC). The rationale of the use of eGE is to differentiate acute from chronic myocarditis. However, recent studies showed a lack of performance of eGE and even combined LLC for diagnosis of acute myocarditis, particularly when compared with native T1 or an extracellular volume fraction evaluation [21]. Thus, we believe that there is a great need for an update of the classical LLC, including CMR mapping techniques.

Finally, the lack of clear clarification of the composition of persistent LGE at one year remains a limitation (fibrosis or inflammatory infiltrates), and an endomyocardial biopsy could have been relevant from this perspective.

Some patients were treated with anti-inflammatory drugs, but it concerned the patients included at the start of the cohort, especially in the case of associated pericarditis. However, animal models recently showed that they can promote viral replication. Today, the patients included in the study do not have a prescription for anti-inflammatory drugs.

## 5. Conclusions

Less than 50% of infarct-like acute myocarditis cases have healed completely at one year, thus highlighting the unfavorable potential of this disease. The initial LGE extent and peak troponin appear to be the only predictors of the absence of complete recovery at one year. Moreover, ventricular dysrhythmia at one year is more frequent in patients with persistent myocardial scarring and we observed a cardiac death before the one-year CMR control in a patient who had precisely persistent myocarditis at three months. This study highlights the importance of maintaining long-term follow-up in these patients, most of whom have normal left ventricular function. Other long-term studies are needed to evaluate cardiovascular events in patients with or without complete recovery.

## Figures and Tables

**Figure 1 jcm-10-04677-f001:**
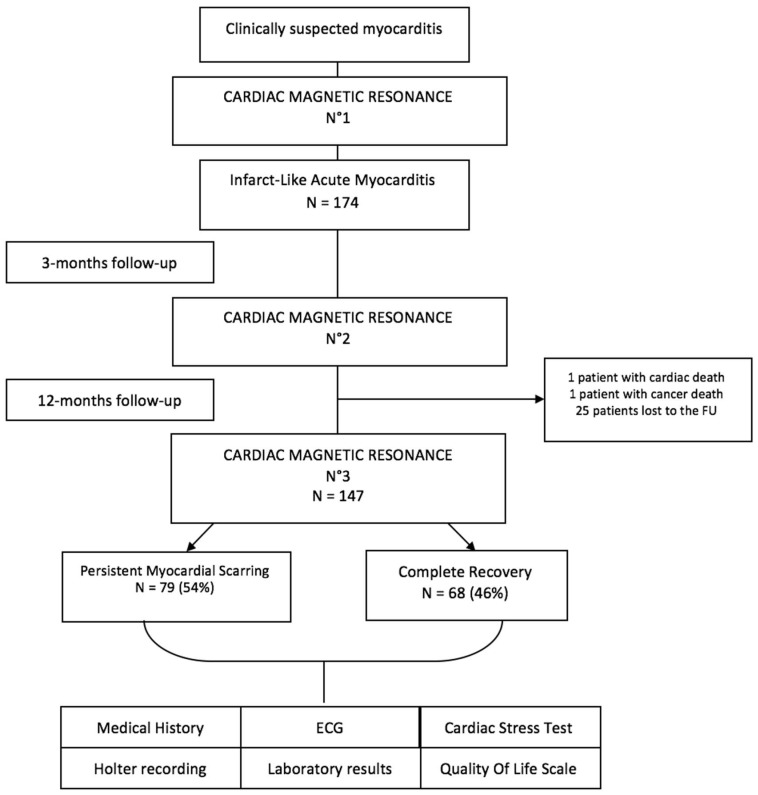
Study flow chart.

**Figure 2 jcm-10-04677-f002:**
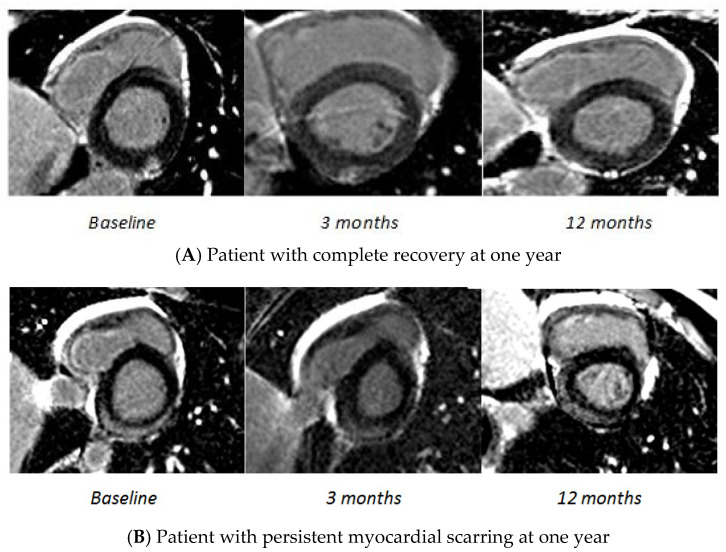
Evolution of late gadolinium enhancement during the follow-up. Cardiovascular magnetic resonance images of LGE. (**A**) Late gadolinium enhancement images demonstrating enhancement in the inferior wall. Complete recovery at one year. (**B**) Late gadolinium enhancement images demonstrating enhancement in the inferior and lateral walls. Persistent impairment at one year.

**Figure 3 jcm-10-04677-f003:**
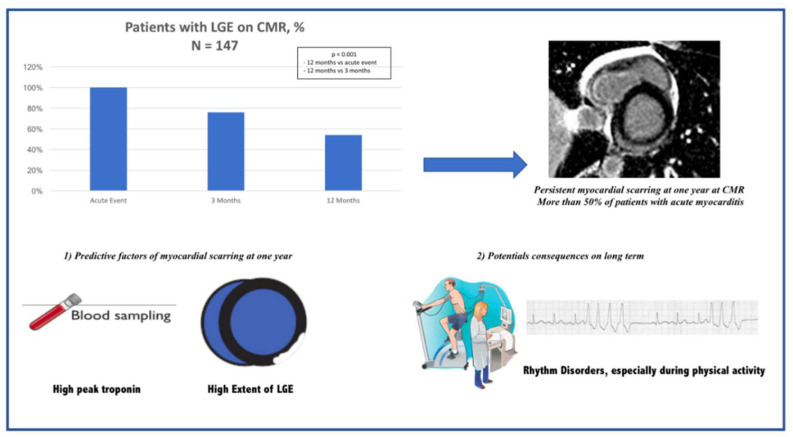
Central illustration.

**Table 1 jcm-10-04677-t001:** Patients’ baseline characteristics.

	*N* = 174
Demographic data	
Age, years	39 ± 17
Males	123 (71%)
Smoking	67 (40%)
Hypertension	26 (15%)
Diabetes	6 (4%)
Hyperlipidemia	17 (10%)
Family history of CAD	22 (13%)
Overweight—Obesity	59 (35%)
Prior autoimmune disease	14 (8%)
Clinical parameters	
Chest pain	174 (100%)
Dyspnea	25 (15%)
Palpitations	11 (7%)
Recent viral history	99 (59%)
ECG—TTE—Coronary artery angiography	
ECG abnormalities	118 (72%)
Repolarization abnormalities	109 (66%)
LVEF at echocardiography, %	60 (55–65)
Coronary artery angiography performed	83 (50%)
Laboratory tests	
Inflammatory syndrome (CRP > 5 mg/L)	134 (83%)
Troponin Ic, peak (ng/mL)	7.4 (2.2–12.0)
Elevated troponin Ic (cTnI > 0.1 ng/mL)	161 (93%)
NT-pro BNP, peak (pg/mL)	433 (172–968)
Medical prescription	
Beta blocker	148 (89%)
ACE inhibitor	151 (92%)
Aspirin (anti-inflammatory dose)	63 (38%)

Values are *n* (%), mean ± SD or median (interquartile range). The data refer to the entire population of 85 patients. CAD = coronary artery disease; CRP = C-reactive protein; ECG = electrocardiogram; TTE = transthoracic echocardiography; LVEF = left ventricle ejection fraction.

**Table 2 jcm-10-04677-t002:** CMR parameters.

	Baseline	3 Months	12 Months	*p*
LVEF, %	57 (52–62)	59 (56–64)	60 (55–65)	
LV dysfunction (<50%)	27 (16%)	10 (6%)	10 (7%)	
Wall motion abnormalities	32 (19%)	10 (6%)	10 (7%)	
LVEDVi, mL/m²	83 (67–96)	81 (67–92)	77 (67–92)	
LVESVi, mL/m²	36 (28–43)	33 (26–39)	32 (25–39)	
LV mass index, g/m²	73 (64–86)	72 (61–82)	70 (58–80)	
LGE present	174 (100%)	127 (76%) *	79 (54%) * ^$^	<0.001
LGE extent (%)	7.35 (4.41–12.5)	2.94 (1.47–5.88) *	1.47 (0.00–4.41) * ^$^	<0.001
No. of segments with LGE	3 ± 2	2 ± 2	1 ± 2	
Edema on CMR	71 (58%)	7 (5%) *	3 (4%) * ^$^	<0.001
Pericardial effusion	30 (19%)	6 (5%)	5 (3%)	
LV dilation	32 (19%)	23 (14%)	22 (15%)	
Predominant Location				
Infero-lateral	133 (77%)	139 (83%)	132 (89%)	
Antero-septo-apical	14 (8%)	12 (7%)	9 (6%)	
Diffuse	27 (15%)	16 (10%)	8 (5%)	

Values are *n* (%), mean ± SD, or median (interquartile range). The data refer to the entire population of 85 patients. CMR = cardiovascular magnetic resonance; LVEF = left ventricle ejection fraction; LV = left ventricle; LVEDVi = left ventricle end-diastolic volume index; LVESVi = left ventricle end-systolic volume index; LGE = late gadolinium enhancement. *p* < 0.001 for: * vs. CMR 1 ^$^ vs. CMR 2.

**Table 3 jcm-10-04677-t003:** Patients’ baseline characteristics according to the presence or absence of LGE at one year (patients with CMR at one year).

	Complete Recovery (*N* = 68)	Persistent Myocarditis (*N* = 79)	*p*
Demographic data			
Age, years	40 ± 18	39 ± 16	0.848
Males	45 (66%)	59 (75%)	0.258
Smoking	27 (42%)	27 (35%)	0.355
Hypertension	12 (19%)	12 (15%)	0.594
Diabetes	3 (5%)	1 (1%)	0.327
Hyperlipidemia	4 (6%)	10 (13%)	0.191
Family history of CAD	7(11%)	13 (17%)	0.329
Overweight—Obesity	19 (30%)	29 (37%)	0.348
Prior autoimmune disease	5 (8%)	6 (8%)	1
Clinical parameters			
Chest pain	66 (100%)	77 (99%)	1
Dyspnea	11 (17%)	11 (14%)	0.652
Palpitations	4 (6%)	7 (9%)	0.529
Recent viral history	33 (51%)	47 (60%)	0.255
ECG and TTE			
ECG abnormalities	43 (71%)	59 (76%)	0.495
Sinus tachycardia	5 (8%)	0 (0%)	0.015
Sustained VT	1 (2%)	0 (0%)	0.439
Repolarization abnormalities	39 (63%)	54 (69%)	0.431
LVEF at echocardiography	60 (60–65)	60 (56–65)	0.197
Laboratory tests			
Inflammatory syndrome (CRP > 5 mg/L)	46 (77%)	61 (80%)	0.611
Troponin Ic, peak (ng/mL)	4.4 (1.4–8.0)	9.2 (3.9–17.5)	<0.001
Elevated troponin Ic	52 (90%)	74 (96%)	0.172
GFR (mL/min/1.73 m^2^)	106.4 ± 29.6	107.5 ± 22.9	0.808
NT-pro BNP, peak (pg/mL)	456 (218–1000)	432 (181–954)	0.931
Medical prescription			
Beta blocker	58 (91%)	67 (87%)	0.501
ACE inhibitor	59 (94%)	69 (90%)	0.396
Aspirin (anti-inflammatory dose)	22 (34%)	35 (45%)	0.204

Values are *n* (%), mean ± SD, or median (interquartile range). The data refer to the entire population of 85 patients. CAD = coronary artery disease; CRP = C-reactive protein; ECG = electrocardiogram; TTE = transthoracic echocardiography; LVEF = left ventricle ejection fraction; CRP = C-reactive protein; GFR = glomerular filtration rate.

**Table 4 jcm-10-04677-t004:** CMR parameters according to the presence or absence of LGE at one year (patients with CMR at one year).

CMR	Baseline		
	Complete Recovery (*N* = 68)	Persistent Myocarditis (*N* = 79)	*p*
LVEF, %	57 ± 9	56 ± 8	0.238
LV dysfunction	6 (9%)	15 (19%)	0.092
Wall motion abnormalities	7 (10%)	16 (20%)	0.122
LVEDVi, mL/m²	80 ± 21	86 ± 19	0.049
LVESVi, mL/m²	31 (23–41)	40 (31–45)	0.001
LV dilation	11 (17%)	15 (20%)	0.727
LV mass index, g/m²	72 ± 15	76 ± 18	0.140
Predominant location LGE			
Infero-lateral	50 (76%)	62 (79%)	0.697
Antero-septo-apical	9 (14%)	3 (4%)	0.032
Diffuse	7 (11%)	14 (18%)	0.225
LGE present	68 (100%)	79 (100%)	1
LGE extent (%)	5.88 (2.94–8.82)	8.82 (5.88–17.65)	<0.001
No. of segments with LGE	2 (1–4)	3 (2–5)	0.004
Edema on CMR	23 (51%)	37/57 (65%)	0.160
Pericardial effusion	15 (25%)	14/70 (20%)	0.462
Global native T1, ms	1046 (1017–1098)	1105 (1052–1150)	0.019
Extra cellular volume	0.29 ± 0.04	0.33 ± 0.05	0.038

Values are *n* (%), mean ± SD, or median (interquartile range). The data refer to the entire population of 85 patients. CMR = cardiovascular magnetic resonance; LGE = late gadolinium enhancement; LVEF = left ventricle ejection fraction; LV = left ventricle; LVEDVi = left ventricle end-diastolic volume index; LVESVi = left ventricle end-systolic volume index.

**Table 5 jcm-10-04677-t005:** Predictors of persistent myocarditis at one year: univariable and multivariate logistic regression analysis.

	Univariable		Multivariate	
	OR (95% CI)	*p*	OR (95% CI)	*p*
Age, years	0.998 (0.979–1.018)	0.846	1.016 (0.992–1.040)	0.199
Males	1.508 (0.738–3.079)	0.260	0.753 (0.308–1.840)	0.534
Troponin Ic, peak > 10 ng/mL	4.305 (1.856–9.983)	0.001	3.032 (1.155–7.964)	0.024
LVEDVi, mL/m	1.017 (1–1.035)	0.052	1.015 (0.995–1.035)	0.155
LGE extent (%)	1.123 (1.056–1.194)	<0.001	1.105 (1.029–1.187)	0.006

ECG = electrocardiogram; GFR = glomerular filtration rate; LVEDVi = left ventricle end-diastolic volume index; LGE = late gadolinium enhancement.

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
