# Peer review of "More than 50% of Persistent Myocardial Scarring at One Year in “Infarct-like” Acute Myocarditis Evaluated by CMR"

_jcm, 2021, doi:10.3390/jcm10204677_

Round 1
Reviewer 1 Report
In this article Pommiera et al addressed an interesting subject., but there are some pitfalls in their analysis.
Major comments
However, the main result of their manuscript, the persistence of limited scarring one year after the diagnosis of acute myocarditis does not seem to be of particular clinical interest. First of all, the prognostic role of late gadolinium enhancement (LGE) regarding future arrhythmic events has been invistigated in patients with some kind of cardiomyopathy, ie patients with an LVEF<50%, thus extrapolating its role to patients with preserved systolic function is a bit arbitary. In this analysis only 10 patients presented such an LVEF at 1 year. Moreover in large prospective trials it has been found that the amount of LGE is proportional to the expected risk. In this article the reported extend of LGE% at one year was 1.47 (0-4.4) -probably reffering to the whole cohort- which seems to be very low.
These concerns seem to be justified by the clinical follow up of these patients. Only one CV death was reported and no significant clinical differences between the two groups were found, except the frequency of PVCs in stress echo which of uncertain clinical significance.
Minor comments
The complete absence of coronary angiography for the diagnosis of these patients could have led both to the inclusion of patients with NSTEMIs in this analysis and to the undertreatment of such patients.
Furthermore, there is no consensus on the use of aspirin or other anti-inflammatory agents in the treatment of myocarditis.
Reviewer 2 Report
This reviewer will congratulate the authors on this nice study. This reviewer thinks that the issues mentioned will improve the manuscript:
General comments:
- The term ”infarct-like” acute myocarditis is misleading – please rephrase. How would a “non-infarct-like” acute myocarditis look like? This reviewer prefers the term: “acute myocarditis” as the authors also use in the Keywords
Specific comments:
Abstract:
- It is stated that MR was performed within 7 days of symptoms – however in the results section it is stated: “The baseline CMR examination was performed 3 ±5 days….”.
- Furthermore it is stated that MR was performed after 3 months follow-up – but no data is displayed within this study – please supply with data from 3 months. Is the same amount of scar present after 3 months of follow-up?
- Second last line in the abstract: …..events were rares,…..” should be changed to “….events were rare,….”.
Methods:
- Why was the “all or none” principal used regarding presence or absence of complete recovery of oedema/scar during follow-up – why was a quantitative or semi-quantitative assessment not used and used as a continuous variable?
- How was premature ventricular contractions defined – 1 beat, 2 beats or other? During stress test it seems to be 5. What do we know about the numbers of PVC in normal control subjects?
- The authors state: ”The extent of myocardial involvement was quantified using a simplified visual quantitative score (SQS) of LGE (10) with the 17-segment LV division of the American Heart Association, which has already been employed in several studies about myocarditis. Segments with LGE were scored according to the number of quartiles involved (1-25%, 26-50%, 51-75%, and 76 to 100%). CMR results were finally expressed as the percentage of LV myocardium involved calculated by adding the 17 segment scores (LGE extent = n% VG = (addition of 17 segment scores / 68) * 100). Each case was also classified according to the predominant location of LGE: infero-lateral, antero-septo-apical or diffuse.” Please add inter- and intraobserver viability on the visual quantitative score of LGE and the inter- and intraobserver viability of the classification regarding predominant location of LGE.
- The authors claim that they use peak Troponin Ic – please clarify the algorithm used assessing peak – how many samples were collected per patient?
Result:
- Please comment on the relative increased amount of prior autoimmune disease states.
- Line 266 – the sentence: “Our results go in line with these studies, showing more events in the group with persistence of LGE at one year.” This reviewer do not think that the data supports this statement- at least not compared to the events referred to in the mentioned reference.
- Please comment on the normal LVEF in this cohort – is EF a good measure? Please see Løgstrup BB et al. Eur Heart J Cardiovasc Imaging 2016.
- Please comment on the very high use of betablockers and ACE-inhibitors, as this reviewer presume is not recommended in myocarditis with normal EF.
- Table 2. This reviewer do not understand the legends used e.g. *$?
- Please add the mean time (with intervals) to CMR investigation after 3 months. Twelve months is provided.
- Of 174 patients included - 147 patients had a 12 months follow-up. Therefore 27 patients “lost to follow-up” at 12 months - what was the reason for this? Was there a selection bias (at least in the 25 patients) – due to selection e.g. patient had a total normal CMR at 3 months no need for further investigation? If this is the case – the title: “More than 50% of persistent myocardial scarring at one year in "infarct-like" acute myocarditis evaluated by CMR” is misleading. Please include these as normals in the different calculations in a sensitivity analysis
Discussion:
- Please discuss if the authors think that it is necessary to perform CMR after 12 months – is it not sufficient to perform it after 3 months? There is no change in LV functional parameters between 3 and 12 months – is LVEF a sufficient measure? – please see my previous comment regarding this issue
Round 2
Reviewer 1 Report
I would like to thank the authors for the time they spent to revise their manuscript based on our comments. Nevertheless – as I reported in my previous review- the design of this study has some inherent limitations. Probably if the follow up was longer, to permit the detection of more events/ or to safely exclude them, these findings would be of greater importance. According to the present analysis I cannot find a reason to prescribe a new CMR at 1 year in a patient with previous myocarditis, normal LVEF with low levels of LGE enhancement. Probably further follow- up of this cohort will provide more robust clinical findings.
Author Response
I share this comment.
A 5-year follow-up is scheduled for this study, with probably more robust results. We thank you for you reviewing.
Reviewer 2 Report
The authors have sufficiently answered this reviewers concerns.
Author Response
We thank you for your reviewing.